# The HIV Restriction Factor Profile in the Brain Is Associated with the Clinical Status and Viral Quantities

**DOI:** 10.3390/v15020316

**Published:** 2023-01-23

**Authors:** Nazanin Mohammadzadeh, Na Zhang, William G. Branton, Ouafa Zghidi-Abouzid, Eric A. Cohen, Benjamin B. Gelman, Jerome Estaquier, Linglong Kong, Christopher Power

**Affiliations:** 1Department of Medical Microbiology and Immunology, University of Alberta, Edmonton, AB T6G 2R3, Canada; 2Department of Mathematical and Statistical Sciences, University of Alberta, Edmonton, AB T6G 2R3, Canada; 3Department of Medicine (Neurology) University of Alberta, 6-11 Heritage Medical Research Centre, Edmonton, AB T6G 2R3, Canada; 4Department of Microbiology and Immunology, CHU de Québec-Université Laval Research Center, Québec, QC G1V 4G2, Canada; 5Institut de Recherches Cliniques de Montreal and Department of Microbiology, Infectiology and Immunology, Université de Montréal, Montreal, QC J2S 2M2, Canada; 6Department of Pathology, University of Texas Medical Branch, Galveston, TX 77555, USA

**Keywords:** host restriction factors, HIV-1, SIV, RNA-seq, transcriptomics, machine learning, ART, *MAN1B1*

## Abstract

HIV-encoded DNA, RNA and proteins persist in the brain despite effective antiretroviral therapy (ART), with undetectable plasma and cerebrospinal fluid viral RNA levels, often in association with neurocognitive impairments. Although the determinants of HIV persistence have garnered attention, the expression and regulation of antiretroviral host restriction factors (RFs) in the brain for HIV and SIV remain unknown. We investigated the transcriptomic profile of antiretroviral RF genes by RNA-sequencing with confirmation by qRT-PCR in the cerebral cortex of people who are uninfected (HIV[−]), those who are HIV-infected without pre-mortem brain disease (HIV[+]), those who are HIV-infected with neurocognitive disorders (HIV[+]/HAND) and those with neurocognitive disorders with encephalitis (HIV[+]/HIVE). We observed significant increases in RF expression in the brains of HIV[+]/HIVE in association with the brain viral load. Machine learning techniques identified *MAN1B1* as a key gene that distinguished the HIV[+] group from the HIV[+] groups with HAND. Analyses of SIV-associated RFs in brains from SIV-infected Chinese rhesus macaques with different ART regimens revealed diminished RF expression among ART-exposed SIV-infected animals, although ART interruption resulted in an induced expression of several RF genes including *OAS3*, *RNASEL*, *MX2* and *MAN1B1*. Thus, the brain displays a distinct expression profile of RFs that is associated with the neurological status as well as the brain viral burden. Moreover, ART interruption can influence the brain’s RF profile, which might contribute to disease outcomes.

## 1. Introduction

Human immunodeficiency virus type 1 (HIV) infects the brain early during systemic infection in cell-free form or via the trafficking of infected cells ([1,2], and reviewed in [3,4]). In the brain, HIV infects perivascular macrophages, microglia, CD4+ T-cells and, to a lesser extent, astrocytes [5,6,7,8,9,10]. Previous reports from our group and others have detected HIV and SIV RNA, DNA and integrated DNA as well as viral capsid proteins in the brain despite long-term suppression with antiretroviral therapy (ART) and undetectable viral RNA in plasma and cerebrospinal fluid (CSF) [11,12,13,14,15]. Although the latent/active status of this reservoir is unknown, the persistence of such a viral reservoir might exert pathogenic effects. This could lead to neurological disorders such as HIV-associated neurocognitive disorders (HAND), seizures and mood disorders [16,17] and contribute to rebound HIV infection in the circulation after the cessation of therapy or the emergence of drug-resistant mutations in HIV, all posing potential hurdles to curing HIV [18].

The unique structure of the brain, with its protected status, makes this anatomical site an especially difficult target for HIV suppression and eradication [19,20,21]. The brain’s organ-specific structural obstacles include the low penetration of ART-associated drugs across the blood–brain barrier (BBB) and the reduced efficacy of ART in limiting the HIV infection of microglia compared to blood-derived lymphocytes ([11,12], and reviewed in [22]), which should be taken into account for other therapeutic interventions including broadly neutralizing antibodies (bNAbs) and CAR T-cells that have yet to be proven to be effective strategies for deep tissues such as the brain [23].

A key component of HAND is the associated neuroinflammation, which is likely driven by the chronic presence of the HIV genome and proteins in the brain [24,25,26,27,28,29]. The proportion of people suffering from HAND has remained unchanged before and after the ART era (~20–30%), but the severity of the neurocognitive impairments has diminished from HIV-associated encephalopathy (HIVE) or dementia (HAD) to the predominance of less severe asymptomatic neurocognitive impairment (ANI) and mild neurocognitive disorder (MND) phenotypes [4,16,30,31,32]. The diagnosis of HAND relies on neuropsychological testing and functional status assessments [4,33] although the presence of comorbidities such as age-related cognitive decline and related systemic diseases as well as the improved recognition of pre-existing mental disorders make the diagnosis of MND and ANI increasingly challenging [34,35]. Moreover, the determinants or a reliable diagnostic biomarker for HAND and the mechanisms underpinning HIV persistence in the brain, together with the impact of this reservoir on the brain’s microenvironment, remain uncertain.

The investigation of host–viral interactions and anti-retroviral restriction factors has yielded substantial insights into systemic HIV pathogenesis (reviewed in [36,37]). Thus, knowledge of these variables in the brain could advance the understanding and treatment of HAND while also highlighting potential biomarkers for clinical diagnosis and monitoring responses to therapy. Antiviral restriction factors are host cellular proteins that play a crucial role in intrinsic immunity and protect the host against pathogenic viruses [37]. Restriction factors (RFs) are usually upregulated by type 1 interferon stimulation during viral infections and target specific components of a virus’s replication cycle directly [38] or act as viral sensors for innate immunity and indirectly restrict the infection [39,40]. These antiviral factors may have co-evolved with the virus and represent signatures of positive selection [41]. Although the presence of specialized RFs in numerous species suggests that they have a higher benefit-to-risk ratio, their constitutive expression during chronic infections can act as a double-edged sword, causing tissue injury while also promoting viral infections [42,43,44,45]. 

The profile of antiviral RFs in the brains of persons with HIV (PWH) and the association of RF expression and neurocognitive impairments are unknown. Herein, we analyzed the transcriptomic profile of anti-retroviral RFs in four clinical groups within a human cohort comprised of persons who were HIV-uninfected (HIV[−]), HIV-infected without pre-mortem neurocognitive impairments (HIV[+]) or HIV-infected with HAND in the presence (HIV[+]/HIVE) or absence of encephalitis (HIV[+]/HAND). Our studies showed that the majority of established HIV-associated RFs were differentially expressed in the brains of HIV[+] patients, and a select group of these factors was highly expressed in HIV[+]/HIVE brains and correlated with neurocognitive impairment, the concomitant higher brain HIV genome and protein levels. However, the causal significance of this association remains unclear. From thirty antiretroviral RFs that we analyzed by quantitative RT-PCR, the *MAN1B1*, *IFITM1* and *IFITM2* gene expression levels were higher in both the HIV[+]/HAND and HIV[+]/HIVE groups compared to the HIV[+] group. The shrinkage discriminant analysis machine learning technique identified *MAN1B1* expression as the most robust distinguishing variable for the HAND diagnosis. To verify these latter findings, the impact of continuous and/or interrupted ART on RF expression in brains was analyzed in a SIV-infected Chinese rhesus macaque model in the presence or absence of effective ART, revealing that *RNASEL* and *MX2* levels were significantly higher in the ART-interrupted group compared to the untreated and treated groups. The SIV-infected ART-treated macaques had lower RF expression levels in the brain compared to untreated and ART-interrupted groups, which might reflect the impact of viral resurgence in CSF and plasma on the brain’s immune response. 

## 2. Materials and Methods

### 2.1. Ethics Statement

The use of autopsied human brain tissues was approved (Pro0002291) by the University of Alberta Human Research Ethics Board (Biomedical), and written informed consent was received for all samples. In addition, brain tissue from patients was obtained from the National NeuroAIDS Tissue Consortium (NNTC) collection. Chinese rhesus macaques (*Macaca mulatta*) were housed at Université Laval in accordance with the rules and regulations of the Canadian Council on Animal Care (http://www.ccac.ca (accessed on 1 October 2019)). The present study protocol was approved by the Laval University Animal Protection Committee (project number 106004).

### 2.2. RNA and DNA Extraction from Brain Tissues

RNeasy and DNeasy kits (Qiagen, Germantown, MD, USA) were used, according to the manufacturer’s protocol, to extract the total RNA and DNA from the midfrontal gyrus of the brains of HIV[−] and HIV[+] patients, as previously described [46]. Using the same technique, total RNA and DNA were extracted from the parietal cortex, striatum and cerebellum of SIV-infected macaques, as previously described [47,48].

### 2.3. Gene Expression Analysis Using Quantitative RT-PCR

The total RNA extracted from the brain tissues of human and non-human primate (NHP) cohorts was used for cDNA synthesis using random primers (Sigma-Aldrich, St. Louis, MO, USA) and SuperScript II reverse transcriptase (ThermoFischer Scientific, Waltham, MA, USA), according to the manufacturer’s instructions. Using appropriate primers (Appendix A), human and rhesus macaque immune gene transcripts were quantified and normalized to GAPDH and reported as the fold change relative to the control group (HIV[−]) for the human cohort and the SIV[+]/ART group for the NHP cohort [46].

### 2.4. Droplet Digital PCR (ddPCR) Analyses of HIV and SIV Brain Viral Load

ddPCR was used to quantify HIV-1- and SIV-encoded RNA, total DNA and integrated DNA in brain tissues. For the quantification of total HIV-1 and/or SIV RNA, 5 μL of cDNA was used as a template along with HIV-1 *pol* and/or SIV *pol* primers (Appendix A). For the quantification of total HIV-1 and/or SIV DNA, 300 ng of genomic DNA (gDNA) along with HIV-1 *pol* and/or SIV *pol* primers (Appendix A) were used as previously described [11]. SIV- and HIV-1-integrated DNA were quantified using 300 ng of gDNA, species-specific primers for Alu and virus-specific primers for gag (Appendix A). QX200 ddPCR EvaGreen supermix and the Bio-Rad QX200 droplet digital PCR system were used as per the manufacturer’s protocol and as previously described [49,50]. Briefly, QX200™ Droplet Generation Oil for EvaGreen and a Bio-Rad QX200 droplet generator were used for droplet generation. This was followed by PCR using an S1000 thermal cycler (Bio-Rad, Hercules, CA, USA). The droplets were analyzed using Bio-Rad QuantaSoft™ Analysis Pro (QuantaSoft AP 1.0.596) Software by setting a common fixed fluorescence threshold intensity based on the no-template control, as previously described [11]. Samples were quantified at least in duplicate, and the viral copies/sample are reported as the copies/gram of tissue (30 mg of brain tissue was initially used for the RNA and DNA extraction for each sample). The values reported for SIV-encoded RNA, total DNA and integrated DNA in the brain represented the mean SIV copies/gram of tissue in the parietal cortex, striatum and cerebellum regions because a test of independence (Chi-square, 5.855; *p * > 0.05) showed that there was no association between the anatomic site and SIV quantity, as previously reported [11].

### 2.5. Human Cohort Study Subjects

These samples were derived from the National NeuroAIDS Tissue Consortium. The cause of death for these patients was not available. The samples were collected the ART era involving different ART regimens and adherence levels; some patients were not receiving ART at the time of death [51].

### 2.6. Animal Housing and Care

The animals were fed a standard monkey chow diet supplemented with fruit, vegetables and water. Social and environmental enrichment was provided by the veterinary staff, and the animals’ health was monitored daily. The animals were evaluated clinically and were humanely euthanized using an overdose of barbiturates according to the guidelines of the Veterinary Medical Association, as previously described [48].

### 2.7. Animal Infection and Sample Collection

Chinese rhesus macaques (*n* = 18) that were confirmed to be seronegative for SIV, STLV-1 (simian T leukemia virus type 1), SRV-1 (simian type D retrovirus 1) and herpes B viruses were infected with SIVmac251 at 10 AID50 intravenously. Four days post-infection, 11 out of 18 animals were treated daily with tenofovir (TFV, 20 mg/kg; Gilead, Foster City, CA, USA) and emtricitabine (FTC, 40 mg/kg; Gilead) subcutaneously and raltegravir (RAL, 20 mg/kg; Merck, Kenilworth, NJ, USA) or dolutegravir (DTG, 5 mg/kg; ViiV, London, UK) and ritonavir (RTV, 20 mg/kg; Abbvie, North Chicago, IL, USA) orally, as previously described [48]. From the 11 ART-treated animals, *n*  =  3 received ART for the duration of the study until they had an undetectable plasma viral load and were euthanized at that time (SIV[+]/ART). *n*  =  8 had analytic therapy interruption (SIV[+]/ATI) at different time points after SIV suppression, and the animals were euthanized when they had a detectable plasma viral load. This duration varied from 10 to 28 days, with two animals requiring 159 and 161 days to rebound. Several animals (*n* = 7) did not receive ART (SIV[+]) and were euthanized 38 to 104 days post-infection. The animals were sacrificed at different time points post-infection, and the harvesting of the plasma, CSF and brain was performed immediately after the sacrifice [48].

### 2.8. RNA-Sequencing and Bioinformatics Analysis

Total RNA was extracted from the cortex at the mid-frontal gyrus of the human cohort (HIV[−], HIV[+], HIV[+]/HAND and HIV[+]/HIVE), as described previously [46]. Five samples from each group, for a total of 20 samples, were subjected to deep sequencing. Sequencing and bioinformatic analysis were performed at the Bioinformatics core facility at the Montreal Clinical Research Institute. The raw read quality was assessed with FASTQC v0.11.8 [52]. Adapter removal was performed using Trimmomatic [53]. The reads were aligned to the GRCh38 (release 77) Ensembl reference genome with STAR v2.5.1b [54]. The raw counts were calculated with FeatureCounts v2.1.0 [55] based on the GRCh38 Ensembl reference genome (release 77). Differential expression was evaluated using the DESeq2 v1. R package [56].

The differential expression results were screened for 60 previously recognized antiretroviral restriction factor (RF) genes. The differential expression of the host RF genes between groups was reported as a log2-fold change, an estimate of the fold change between conditions determined from the distribution of the reads.

### 2.9. Statistical Analyses for qRT-PCR and ddPCR

Prism 9.4.1 software (GraphPad Software, San Diego, CA, USA) was used to assess significant differences in gene expression between groups. The outliers were eliminated using the ROUT method, with the ROUT coefficient set to 0.1%, ensuring the removal of only definitive outliers. For multiple comparison analysis, Tukey’s multiple comparisons test was performed after ordinary one-way ANOVA. Asterisks are used to display *p* values. The *p* value threshold was set to 0.05, and the statistical significance was displayed as * for *p* ≤ 0.05, ** for *p* ≤ 0.01, *** for *p* ≤ 0.001, and **** for *p* ≤ 0.0001.

### 2.10. Variable Importance Plots

The variable importance plots are generated using the ensemble machine learning methods, which are advanced techniques often used to handle complex machine learning problems. The basis of ensemble learning is to build a strong classification model by combining the strengths of a collection of weak but simpler classifiers [57,58]. We selected random forest (RF) [59], Shrinkage Discriminant Analysis (SDA) [60], regularized random forest (RRF) [59] and Regularized Logistic Regression [61] as our base classifiers. Instead of aggregating their results, we only specified the model with the best performance, from which we plotted the variable importance. Variable Importance (VI) is a general procedure for selecting interesting covariates in a prediction model, which can be used with any regression and classification method. This procedure generates a measure of importance for each covariate by estimating the response variable with some perturbations of the covariate and computing the error due to these perturbations [62]. The covariates with the highest VI are assumed to be the most important in predicting the response variable. In order to have a robust estimation of the importance of the covariates, the procedure is replicated many times for each covariate. For this analysis, all measures of importance are scaled to have a maximum value of 100; we present the scaled mean VI of each covariate in the variable importance plot and highlight the covariates with a scaled mean VI above a threshold of 70.

## 3. Results

### 3.1. Quantification of the Viral Burden in the Plasma, CSF and Cerebral Cortex of Persons with HIV

The profile of antiretroviral RFs in the brain and how it changes during HAND are unknown. This knowledge gap prompted us to determine the viral load in the plasma, CSF and cerebral cortex from persons who were uninfected (HIV[−], *n* = 10), HIV-infected without neurological disease at death (HIV[+], *n* = 10), HIV-infected with HAND (HIV[+]/HAND, *n* = 10) and HIV-infected with HAND and HIV encephalitis (HIV[+]/HIVE, *n* = 10) (Table 1). The plasma and CSF viral loads as well as the HIV RNA, DNA and integrated DNA in brains from the HIV-uninfected group (HIV[−]) were undetectable (Figure 1). The mean plasma viral RNA levels were similar between the three HIV-infected groups (Figure 1A, HIV[+], 5.1 log RNA copies/mL, HIV[+]/HAND, 5.4 log RNA copies/mL and HIV[+]HIVE, 5.6 log RNA copies/mL). The mean CSF viral RNA level was higher in the HIV[+]/HIVE group (5.9 log RNA copies/mL) compared to that of the HIV[+] (4.1 log RNA copies/mL) (*p* < 0.01) and HIV[+]/HAND (3.1 log RNA copies/mL) (*p* < 0.01) groups (Figure 1A). The mean cerebral cortex HIV RNA levels were higher in HIV[+]/HIVE (5.0 log copies/g) compared to those in HIV[+]/HAND (4.6 log copies/g) (*p* < 0.01) and HIV[+] (3.8 log copies/g) (*p* < 0.0001) (Figure 1B). This trend was similar for the HIV total DNA levels in the brain (Figure 1C). The mean HIV total DNA was higher in HIV[+]/HIVE (5.2 log copies/g) compared to that in HIV[+]/HAND (4.7 log copies/g) (*p* < 0.005) and HIV[+] (4.6 log copies/g) (*p* < 0.005) (Figure 1C). The integrated HIV DNA levels were similar between the three HIV[+] groups (HIV[+], 3.9 log copies/g, HIV[+]/HAND, 4.0 log copies/g and HIV[+]/HIVE, 4.6 log copies/g), without statistically significant differences (Figure 1D).

### 3.2. Identification of Differentially Expressed Host Restriction Factors in HIV-Infected Brains by RNA-Sequencing

To identify the differentially expressed RFs in the brains of PWH, we performed bulk RNA-sequencing using the total RNA derived from the brains of the human cohort (Table 1). After an extensive literature search that examined all ISGs and RFs that were associated with HIV infection, we identified sixty differentially expressed anti-retroviral RFs and reported a log2 fold change (log2 FC) of each gene in various group comparisons (Figure 2 and Appendix A). The comparison the of RFs’ mRNA expression in the brains of PWH compared to that of the control (HIV[−]) revealed an average upregulation of 78% of these genes in HIV[+] brains with and without HAND (Figure 2A–C). From 60 RFs that were identified in the brain, 46 were upregulated in the brains of the HIV[+] group compared to HIV[−], ranging from 0.1 to 3.08 log2 FC (Figure 2A).This was accompanied by the upregulation of 47 genes in the brains of HIV[+]/HAND (0.1 to 3.5 log2 FC) and 49 genes in the brains of HIV[+]/HIVE (0.1 to 3.45 log2 FC) compared to HIV[−] (Figure 2B,C).

The comparison of the RFs’ mRNA expression between the brains of the HIV[+]HAND and HIV[+]/HIVE groups identified an average of 59% upregulated genes (0.1 to 2.4 log2 FC) (Appendix A). We observed 36 out of 60 RFs upregulated in the brains of HIV[+]/HAND compared to the HIV[+] group (0.1 to1.78) (Appendix A). The brains of the HIV[+]/HIVE group had 41 upregulated RFs compared to the HIV[+] group (Appendix A). The comparison of the two HAND groups, HIV[+]/HAND and HIV[+]/HIVE, indicated 30 upregulated genes, with the log2 FC ranging from 0.1 to 2.4 (Appendix A). Overall, the number of upregulated RFs and the magnitude of this upregulation were higher in the comparisons between the HIV-positive groups and the HIV[−] group compared to the comparisons between the HIV[+] groups with and without HAND 

The opposite trend was observed for the downregulated genes. The mean percentage and the magnitude of downregulation (13.7% and −0.1 to −0.7 log2 FC, respectively) were lower in the HIV[+] groups vs. HIV[−] comparisons (Figure 2) than they were in the comparisons between the diseased groups (22.3% and −0.1 to −1.4 log2 FC) (Appendix A). Overall, we observed a limited downregulation of RFs among the group comparisons. 

To focus on the highly induced RFs for further analysis, we selected the top ten RFs that were upregulated in the HIV[+] groups compared to the HIV[−] control (1.77 to 3.5 log2 FC) (Figure 3). To select the top upregulated genes from the HAND group comparisons, since the magnitude of upregulation was modest, as previously mentioned, we chose the RFs with the highest log2 FC ranging from 1 to 2.41 (Figure 3). Members of the OAS, GBP and IFITM families were commonly upregulated between various group comparisons. Of the 60 RFs, 19 unique genes were highly induced after HIV infection and/or HAND, which were selected for further analysis and confirmation (Figure 3). 

### 3.3. Verification of Highly Upregulated RFs by Quantitative Reverse Transcription PCR

Quantitative Reverse Transcription PCR (qRT-PCR) was used to validate the differential expression of the selected genes. The relative mRNA expression of the majority of these genes—*GBP2*, *GBP5*, *IFITM1*, *IFIT1-3*, *CIITA*, *CCL8*, *IFI16*, *OAS1-3*, *OASL*, *RNASEL*, *MX2*, and *ISG15*—was higher in the HIV[+]/HIVE group compared to the HIV[−] and HIV[+] groups (Figure 4). IFITMs (interferon-induced transmembrane proteins) inhibit the entry/fusion of some enveloped viruses, including HIV, by changing the physical characteristics of the plasma membrane of the host cells [63,64]. *IFITM1* was upregulated in both HIV[+] groups with HAND, although this upregulation was statistically significant only in the HIV[+]/HIVE group compared to the HIV[−] (*p* < 0.01) and HIV[+] (*p* < 0.01) groups (Figure 4A). Interestingly, the *IFITM2* gene was only upregulated in the HIV[+]/HAND group compared to the HIV[+] group (*p* < 0.05) (Figure 4B). The *IFITM3* mRNA expression levels did not differ among different groups (Figure 4C).

GBP2 and GBP5 are two members of the Guanylate-binding protein (GBP) family that inhibit a host protease (furin) and reduce the furin-mediated cleavage of cellular and viral proteins, including the HIV envelope (gp160), into gp120 and gp41 [65,66]. Both *GBP2* and *GBP5* were upregulated in the HIV[+]/HIVE group compared to the control (*p* < 0.05 and *p* < 0.05, respectively) (Figure 4D,E).

ISG15 restricts HIV release by inhibiting the ubiquitination of the HIV Gag and host Tsg101 (tumor susceptibility gene 101) proteins and, subsequently, their interaction, both of which are involved in HIV budding [67,68]. *ISG15* was highly upregulated in HIV[+]/HIVE brains compared to HIV[+]/HAND (*p* < 0.01) and HIV[−] (*p* < 0.001) brains (Figure 4F).

Next, we assessed the mRNA expression of the IFIT family. Like IFITMs, IFITs (IFIT1, IFIT2 and IFIT3) are broad-spectrum antiviral restriction factors [69]. Nasr et al. showed that the siRNA knockdown of IFITs in monocyte-derived macrophages increased HIV production [70]. All three members of the IFIT family had robust upregulation in HIV[+]/HIVE brains compared to the HIV[−], HIV[+] and HIV[+]/HAND groups (Figure 4G–I). 

In the same manner, CIITA, CCL8, IFI16 and MX2, which target viral transcription, entry, latency reactivation/transcription and HIV nuclear import steps in HIV replication, respectively [71,72,73,74], showed robust upregulation in the HIV[+]/HIVE group compared to the rest of the cohort (Figure 4J–M). Among these genes, the *CCL8* and *MX2* mRNA levels increased by more than 10-fold in the HIVE group compared to the control (Figure 4K–M). 

All members of the OAS family (*OAS1 to 3* and *OASL*) and the associated *RNASEL* gene were among the top upregulated RFs. OAS proteins are nucleotidyltransferases that activate RNase L. RNase L subsequently degrades viral and cellular RNAs [75]. OAS family gene expression has been associated with HIV-associated neurocognitive disorders [76]. The *OAS1* mRNA expression was 17.8-fold higher in the HIVE group compared to that in the HIV[−] control (*p* < 0.0001) (Figure 4N). *OAS2* showed a 9.3-fold increase in the HIVE group compared to the HIV[−] control (*p* < 0.0001) (Figure 4O), and the *OAS3* mRNA levels were 16-fold higher in the HIVE group compared to those in the control (*p* < 0.0001) (Figure 4P). The *OASL* and *RNASEL* levels were 10.4- and 3.6-fold higher in the HIVE group compared to the HIV[−] control (*p* < 0.0001) (Figure 4Q,R). 

*BST2,* which encodes Tetherin and inhibits the release of budding virions [77], trended higher in the HIV[+] and HIV[+]/HIVE groups; however, this was not statistically significant (Appendix A). The two type-1 interferon genes that we tested, *IFNB1* and *IFNA,* displayed similar expression levels across groups in the cohort (Appendix A).

### 3.4. Verification of Downregulated RFs by Quantitative Reverse Transcription PCR

From the 60 differentially expressed RFs, we selected the genes that were downregulated in different group comparisons (Figure 5). The magnitude of downregulation was restricted and ranged between −0.4 and −1.4 log2 FC. We selected nine RF genes that did not overlap with previously confirmed upregulated genes for qRT-PCR validation. qRT-PCR revealed that these genes were not significantly downregulated (Appendix A); in fact, *SELPLG, PPIA, TREX1* and *MAN1B1* were highly induced in the HIV[+]/HIVE group compared to the HIV[−] group (Figure 6A–D). TREX1 is an exonuclease that degrades cellular and viral cytoplasmic DNAs to prevent autoimmunity and viral infections [78,79]. TREX1 participates in the downregulation of type I interferons and might contribute to HIV latency and persistence [43,79]. Similarly, the *PPIA*-associated protein Human cyclophilin A can increase HIV infectivity by facilitating the uncoating event [80].

Interestingly, *MAN1B1*, which encodes the ERMAN1 protein and exerts a recognized effect on HIV envelope degradation via the endoplasmic reticulum-associated protein degradation pathway [81], was transcriptionally upregulated in both HIV[+] groups with HAND, with statistical significance between the HIV[+]/HIVE and HIV[−] groups (7.7-fold, *p* < 0.005) as well as the HIV[+]/HIVE and HIV[+] groups (3.9-fold, *p* < 0.05) and the HIV[+]/HAND and HIV[−] groups (6-fold, *p* < 0.05) (Figure 6D). The differential mRNA expression of *SERINC3, MARCHF2, CH25H, APOBEC3A* and *APOBEC3B* between groups lacked statistical significance (Appendix A). Notably, some mRNAs were found to be downregulated in RNA-seq but were upregulated in the RT-qPCR analyses. This discrepancy could be due to the limited and group-dependent suppression of downregulated genes and the presence of multiple pseudo-genes that might bias the RNA-seq dataset.

### 3.5. Machine Learning Strategies Identified Key Variables for Classifying HAND Groups

To analyze the distinguishing capacities of individual variables on clinical group classification, including clinical variables (age, sex, plasma/CSF viral loads), host gene expression levels (30 genes verified by qRT-PCR including 19 upregulated genes, 9 downregulated genes and IFNα and IFNβ1) and brain HIV quantities (e.g., RNA, total DNA and integrated DNA), we applied several machine learning algorithms to elucidate which variables were predictive of individual clinical groups. The regularized logistic regression analysis of the HIV[−] and HIV[+] groups relied on the plasma viral RNA, blood CD4+ T-cell count, brain viral DNA and integrated DNA to distinguish HIV[−] from HIV[+] brains (value of importance >70) (Figure 7A). The comparison of the HIV[+] group with the HIV[+]/HAND group using the shrinkage discriminant analysis showed *MAN1B1*, with an importance value of 100, and age, with an importance value of 70, as the two key variables that distinguished these two groups (Figure 7B). The comparison of the other HAND group, HIV[+]/HIVE (with the HIV[+] group), again predicted *MAN1B1*, with an importance value of 100, as a distinguishing variable between these two groups (Appendix A). When all four groups are compared, members of the OAS family (*OAS1,3, L* and the related *RNASEL* gene) were among the most robust variables for group discrimination (Appendix A). The same trend was observed between the two HAND groups (HIV[+]/HAND and HIV[+]/HIVE) (Appendix A).

To investigate the relationships between viral quantities in the brain and concurrent host RF levels, correlational analyses were performed (Appendix A and Appendix A). Among the analyzed genes, *MX2* and *ISG15* showed the highest positive correlation with viral quantities in the brain, while *IFNA* and *IFNB1* were highly negatively correlated with brain viral quantities. *MX2* was significantly and positively correlated with brain viral RNA (coefficient: 0.77; *p* < 0.00001), brain viral DNA (coefficient: 0.84; *p* < 0.00001) and brain viral integrated DNA (coefficient: 0.86; *p* < 0.00001). In the same manner, the ISG15 transcript levels correlated positively with brain viral RNA (coefficient: 0.69; *p* < 0.00001)*,* brain viral DNA (coefficient: 0.76; *p* < 0.00001) and brain viral integrated DNA (coefficient: 0.87; *p* < 0.0001) levels. *IFNA* transcript levels correlated negatively with brain viral RNA (coefficient: −0.59; *p* < 0.0005)*,* brain viral DNA (coefficient: −0.53; *p <* 0.005) and brain viral integrated DNA (coefficient: −0.54; *p* < 0.001)*. IFNB1* levels correlated negatively with brain viral RNA (coefficient: −0.51; *p* < 0.005), brain viral DNA (coefficient: −0.52; *p* < 0.005) and brain viral integrated DNA (coefficient: −0.41; *p* < 0.05). These data underscore the divergent RF and antiviral expression levels in relation to brain viral quantities.

### 3.6. Impact of ART on RFs in the Brains of the Non-Human Primate Cohort

Previous studies have examined the impact of ART on the brain viral load in HIV-infected humans and SIV-infected non-human primates [11,15]. While HIV and SIV genomes and proteins persist in the brain despite ART-mediated suppressed plasma and CSF viral loads, the impact of ART on the expression profile of RFs in the brain is unknown. We examined the effects of contemporary ART drugs (tenofovir, raltegravir/dolutegravir, ritonavir and emtricitabine) on previously identified RFs in brains from SIV-infected Chinese Rhesus macaques. This SIV-infected non-human primate (NHP) model included three groups: (1) SIV-infected and ART naïve (SIV[+]), *n =* 7; (2) SIV-infected with treatment interruption (SIV[+]/ATI), *n =* 8; and (3) SIV-infected with suppressive ART (SIV[+]/ART), *n =* 3 (Table 2). 

Based on the human RF analysis, we performed selective qRT-PCR gene expression analyses in the NHP cohort. Among the *OAS* family genes and *RNASEL*, *OAS1* was upregulated in the SIV[+]/ATI group, although it lacked statistical significance (Figure 8A). The mRNA expression of the *OAS2* gene was similar within the NHP cohort (Figure 8B). However, the *OAS3* mRNA expression levels were higher in both the SIV[+] and SIV[+]/ATI groups compared to those in the ART-treated group (*p* < 0.05 and *p* < 0.05, respectively) (Figure 8C). *RNASEL* displayed robust upregulation in the SIV[+]/ATI group compared to the SIV[+] and SIV[+]/ART groups (*p* < 0.05 and *p* < 0.05, respectively) (Figure 8D). ART and ART interruption did not have any impact on the expression levels of *IFIT3* and *TRIM5α* in the NHP cohort (Figure 8E,F). The levels of *SAMHD1* mRNA were reduced in the SIV[+]/ART group compared to those in the untreated and interrupted groups; however, this lacked statistical significance (Figure 8G). This trend was also observed for *BST2* mRNA levels, which were diminished in the ART-treated group compared to the untreated group (*p* < 0.05) (Figure 8H). The profile of *MX2* mRNA expression resembled *RNASEL*, in which the transcript was upregulated in the ART-interrupted group compared to that in the ART naïve and treated groups (*p* < 0.05 and *p* < 0.05, respectively) (Figure 8I).

Since *MAN1B1* was a pivotal discriminating variable for the HIV[+] and HIV[+] groups with HAND (Figure 6D and Figure 7B), we measured the relative mRNA expression levels of this gene in the NHP cohort to understand the impact of ART on this gene. *MAN1B1* mRNA was upregulated 4.8-fold in the SIV[+] group compared to SIV[+]/ART (*p* < 0.005). This was accompanied by a 4.6-fold upregulation after ART interruption in SIV[+]/ATI (*p* < 0.005) (Figure 8J). We measured *IFNA* and *IFNB1* in the NHP cohort, revealing that the expression level of these two genes was lower in the SIV[+]/ART group compared to that in the other groups, and this difference was statistically significant for *IFNB* in SIV[+]/ART compared to that in SIV[+]/ATI (*p* < 0.05) (Figure 8K,L).

## 4. Discussion

In the present study, we identified how the expression profile of antiretroviral RFs in the brain is affected by HIV infection, the development of HAND and ART exposure. There was a robust upregulation of most RFs in the brains from persons with HIV[+]/HIVE, which was verified by qRT-PCR following RNA-seq. These findings were aligned with high plasma, CSF, brain viral RNA and total DNA levels in this clinical group. The *IFITM1* and *MAN1B1* mRNA expression levels were higher in both HAND groups, while the machine learning algorithms identified *MAN1B1* as an important variable that distinguished both HAND groups from the HIV[+] group without neurocognitive impairments. Moreover, ART exerted a differential impact on brain RFs in the SIV-infected NHP model. Overall, RFs exhibited a diminished expression in the SIV[+]/ART group compared to that in the SIV[+] and SIV[+]/ATI groups. Similarly, *OAS3*, *MAN1B1* and *BST2* were highly upregulated in the SIV[+] group, but the mRNA expression levels diminished with effective ART. 

A previous study identified the upregulation of interferon-stimulated genes (ISGs) in the brains of HIV[+] patients with and without neurocognitive impairments [82]. Our results recapitulate these findings by showing the induction of RFs during HIV infection, especially among persons with HIVE. However, the present studies focus on a subset of ISGs with specialized restrictive properties for lentivirus infections. RFs usually exert broad antiviral activities, and many of the RFs that we screened in the RNA-seq database are reported to inhibit a range of viral infections in addition to their anti-retroviral functions [64,66,83]. A transcriptomic study by Sanfilippo et al. identified the upregulation of all OAS gene family members in brains from SIVE (SIV[+] with encephalitis) animals as well as brains from persons with HAND [76]. In the present study, we observed the robust upregulation of the *OAS* family and the associated *RNASEL* gene in the brains of HIV[+] patients with and without HAND. qRT-PCR further confirmed that these genes are highly expressed in the brains of HIV[+]/HIVE patients, and machine learning methods identified this family as the distinguishing factor between the HIV[+]/HAND and HIVE groups. 

The present manuscript might benefit from a larger sample size in both the human and NHP cohorts. We observed discordance between the qRT-PCR and RNA-sequencing, which was more apparent for the verification of the downregulated RF transcripts. Another limitation was the uncertain ART status within the human cohort. The HIV-infected patients in the human cohort might not have been receiving ART at the time of death, and the precise history of their ART exposure (and specific ART regimen) at the time of death was unavailable. This study analyzed the expression of RFs in the brain in RNA levels and not protein levels. Indeed, the levels of expression of RFs in protein levels, subcellular localization and host protein–protein and viral-host protein–protein interactions should be further investigated using proteomics strategies.

*MAN1B1* might be a potential pathogenic determinant as well as a biomarker for neurocognitive impairments in PWH. *MAN1B1* encoded the protein ERMAN1, which is an endoplasmic reticulum (ER)-associated α-mannosidase. ERMAN1 is constitutively expressed and plays a crucial role in protein folding and processing [84]. Nascent proteins undergo glycosylation upon entry to ER. In this process, fourteen pre-assembled oligosaccharides (two *N*-acetylglucosamine, nine mannoses and three glucose residues) are added to an Asn residue on the precursor protein. The removal of these sugars in different steps dictates the proteins’ fate. The removal of the three glucose residues guides the client protein to interact with ER chaperons, where they fold and oligomerize into native proteins. In the next step, ERMAN1 removes the outer mannose residue from the preassembled oligosaccharide tag that is N-linked to the native protein; this signals for the release of the protein from ER to its final destination [44,84,85]. However, if the proteins are not properly folded, ERMAN1 continues the cleavage of mannoses beyond the outer residue and to the terminal mannose units, sorting the misfolded protein for the endoplasmic reticulum-associated degradation (ERAD) pathway [86]. 

The HIV env protein gp160 enters ER for proper folding and oligomerization into a functional trimer. ERMAN1 selectively interacts with HIV env gp160 in ER via its catalytic domain and processes subsequent HIV env degradation by ERAD [81]. ERMAN1 has been associated with the pathogenesis of several conditions. ERMAN1 overexpression is associated with a poor prognosis of bladder cancer [87]. It has been shown that ERMAN1 can increase the proliferation, migration and invasion of hepatoma cells in hepatocellular carcinoma [88]. *MAN1B1* deficiency is linked to Congenital Disorders of Glycosylation (CDG) that can manifest with intellectual and developmental disabilities [89]. ERMAN1 overexpression can result in the increased demannosylation of ER-associated proteins [90,91] and a subsequent increase in protein degradation via ERAD and/or glycosylation challenges that may lead to protein dysfunction and cell/tissue damage.

The expression profile of *MAN1B1* and its associated protein in the CSF and plasma of PWH with and without HAND are unknown and warrant further investigation as potential biomarkers. Our future studies will examine the protein expression levels of ERMAN1 in brains and CSF from PWH. Further studies of *MAN1B1* expression in the brain and other tissues might elucidate the effects of this gene and whether it and associated pathways exert cell injury in HAND, which can be targeted for therapeutic intervention.

In conclusion, using a high-throughput strategy coupled with molecular verification and machine learning tools, we report that a select group of RFs are induced in the brains of HIV[+] groups with HAND, which should be further investigated for their potential clinical applications. Moreover, we showed that ART exerted effects on the steady-state expression of RFs in the brain, although ART interruption increased the expression level of these same genes, indicating the importance of ongoing adherence to ART, monitoring viral blips and, accordingly, the modulation of ART regimens.

## Figures and Tables

**Figure 1 viruses-15-00316-f001:**
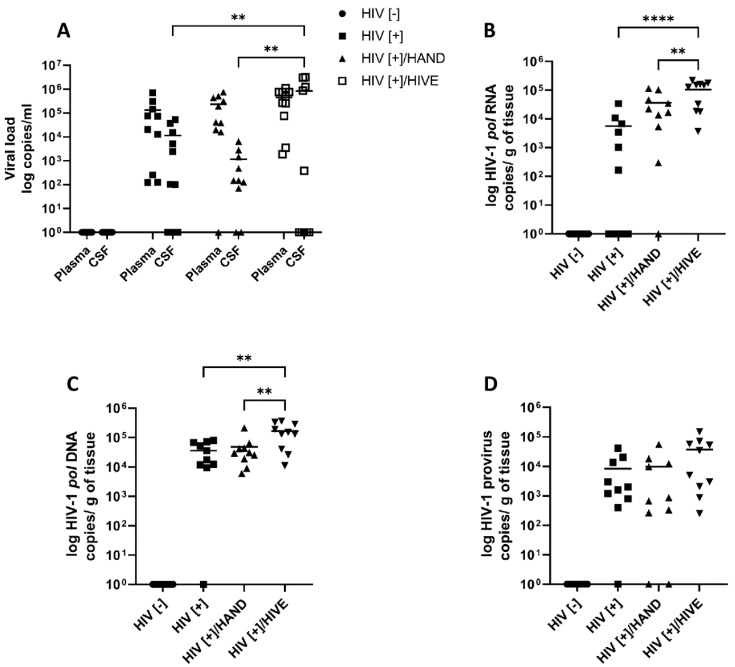
**Plasma, CSF and brain viral quantities in the human cohort**. Four experimental groups ((HIV[−], *n*  =  10), (HIV[+], *n*  =  10), (HIV[+]/HAND, *n*  =  10) and (HIV[+]/HIVE, *n*  =  10)) were examined for plasma and CSF HIV RNA copies/mL (**A**), HIV RNA (**B**), total DNA (**C**) and integrated DNA (**D**) copies/g of brain tissue. The horizontal lines represent the mean values *(***, *p* < 0.01; ******, *p* < 0.0001).

**Figure 2 viruses-15-00316-f002:**
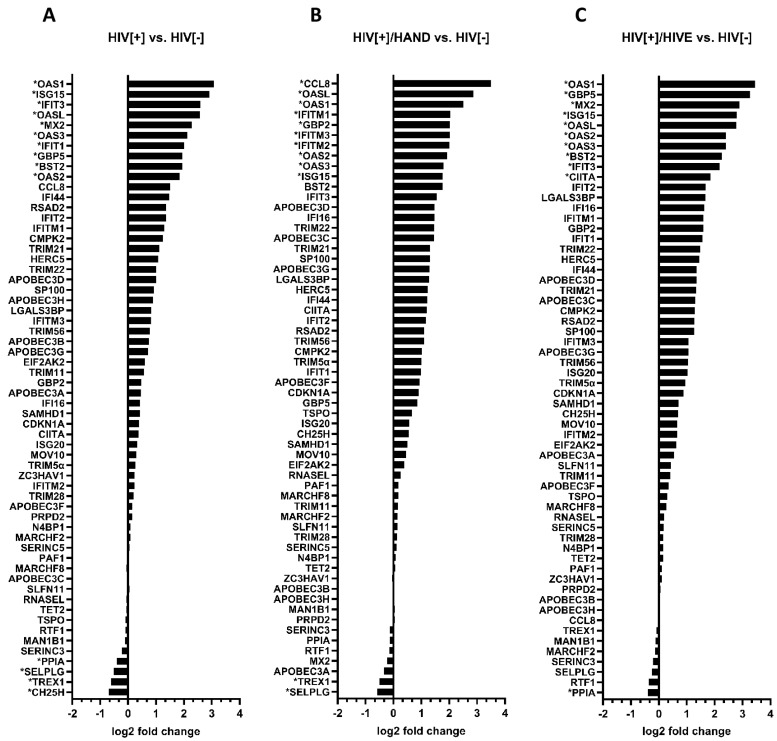
**Relative host restriction factors’ mRNA expression levels.** RNA isolated from the post-mortem brains of the human cohort was subjected to bulk RNA-seq. Sixty known restriction factor genes with HIV restrictive capabilities were screened in the RNA-seq dataset. Differentially expressed genes are reported in different group comparisons. (**A**) HIV[+] compared to HIV[−], (**B**) HIV[+]/HAND compared to HIV[−] and (**C**) HIV[+]/HIVE compared to HIV[−]. The fold changes range from −2 to −1, 0, 1, 2, 3 and 4 log2 fold changes (or 0.25, 0.5, 1, 2, 4, 8 and 16-fold changes). Genes with 0 to −2 log2 fold changes are considered downregulated, and those with 0 to 4 log2 fold changes are considered upregulated. Asterisks indicate the genes that were selected for validation by qRT-PCR.

**Figure 3 viruses-15-00316-f003:**
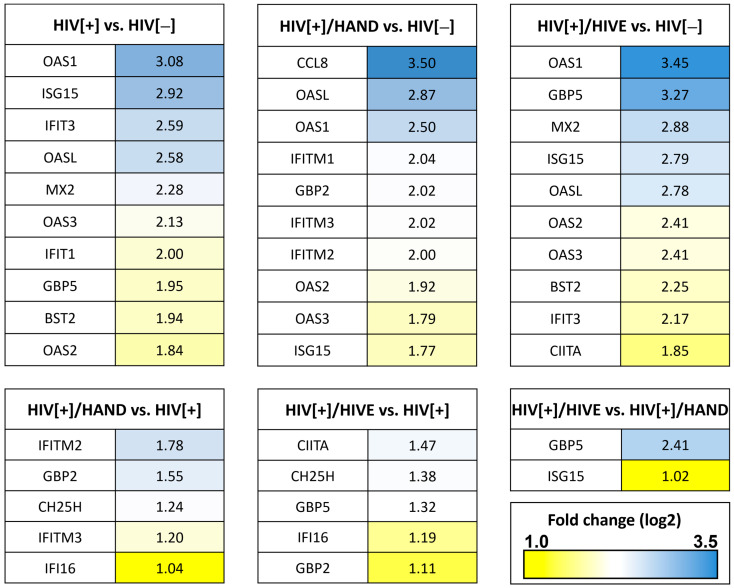
**Heat map of upregulated host restriction factors in different experimental groups.** The top upregulated genes (*n* = 10) were selected from each of the HIV[+] groups compared to the HIV[−] group derived from the RNA-seq database. For the comparisons within HIV[+] groups, with or without HAND, the top upregulated genes with a minimum 1.0 log2 fold change were selected.

**Figure 4 viruses-15-00316-f004:**
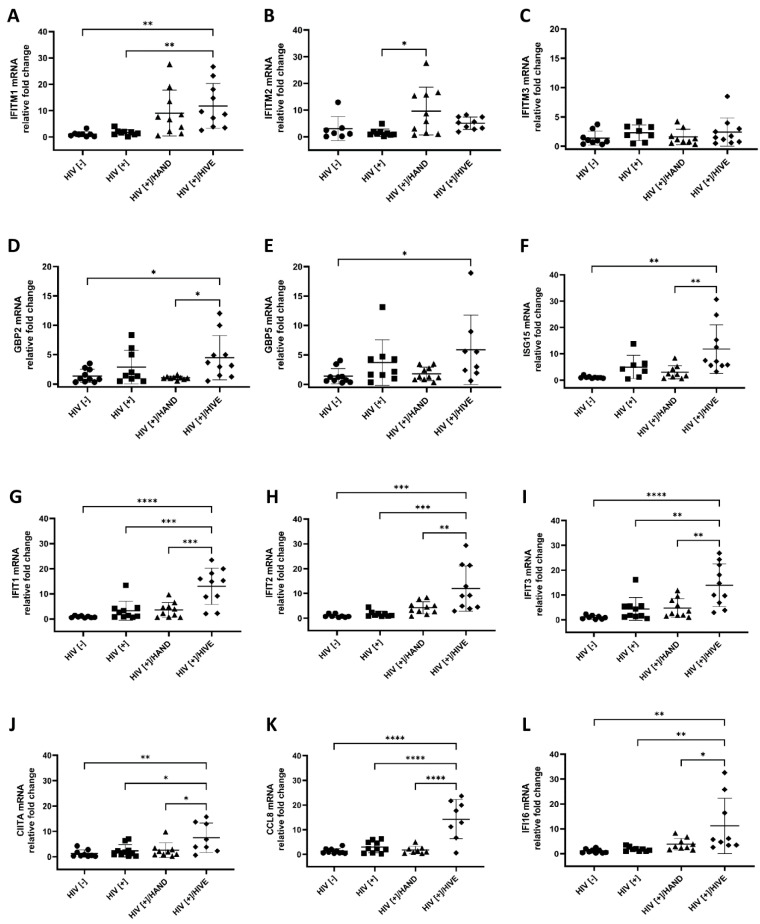
**Relative host restriction factors’ mRNA expression levels**. Highly upregulated RFs identified in the RNA-seq database were subjected to qRT-PCR validation. The mRNA expression levels for each gene (**A**–**R**) were normalized to GAPDH and are reported as the fold change relative to the HIV[−] control group. The horizontal lines represent the mean values and the error bars represent the standard deviation *(**, *p* < 0.05; **, *p* < 0.01; ***, *p* < 0.001; ******, *p* < 0.0001).

**Figure 5 viruses-15-00316-f005:**
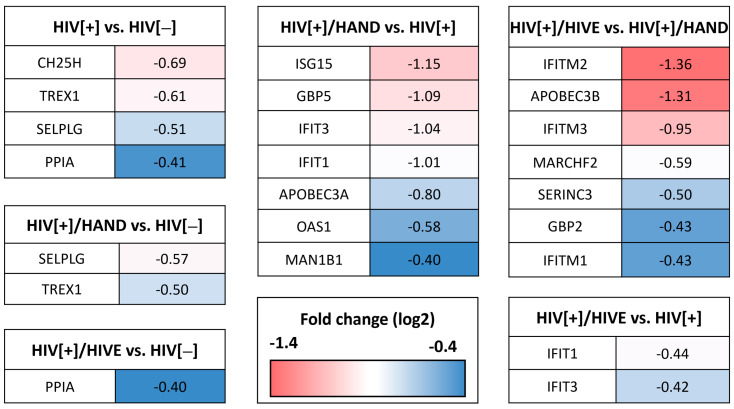
**Comparison of downregulated restriction factors in different experimental groups.** Restriction factors with a minimum −0.4 log2 fold change (0.75 fold change) were selected from different group comparisons based on the RNA-seq dataset. The most downregulated genes belonged to the two HIV[+] groups with HAND, where *IFITM2* and *APOBEC3B* had −1.36 and −1.31 log2 fold change downregulation in the HIV[+]/HAND group compared to the HIV[+]/HIVE group, respectively.

**Figure 6 viruses-15-00316-f006:**
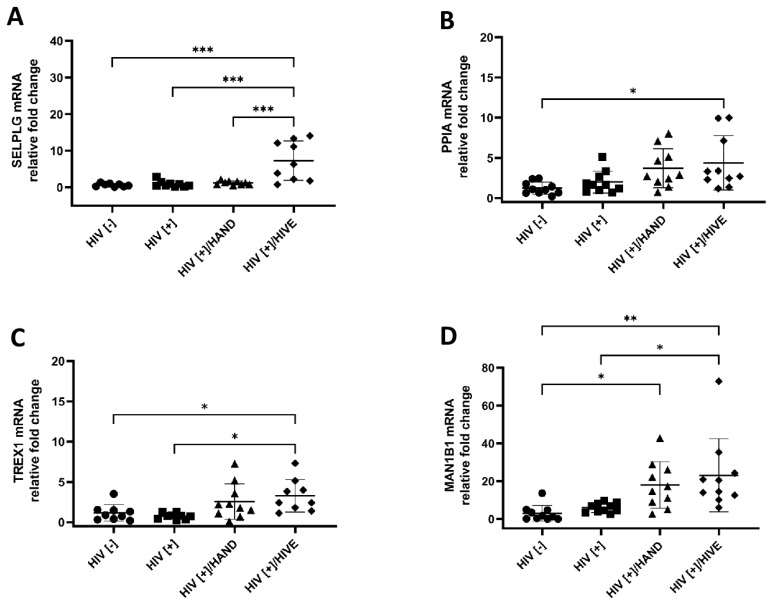
**Relative mRNA expression of downregulated restriction factors.** The top downregulated genes validated by qRT-PCR using appropriate primers showed that the *SELPLG, PPIA* and *TREX1* mRNA expression levels were not lower in the HIV[+] and HIV[+]/HAND groups compared to HIV[−] (**A**–**C**). The *MAN1B1* gene was upregulated in both HAND groups compared to the HIV[+] and HIV[−] groups, despite the RNA-seq suggesting otherwise (**D**). The horizontal lines represent mean values, and the error bars represent the standard deviation (*, *p* < 0.05; **, *p* < 0.01; ***, *p* < 0.001).

**Figure 7 viruses-15-00316-f007:**
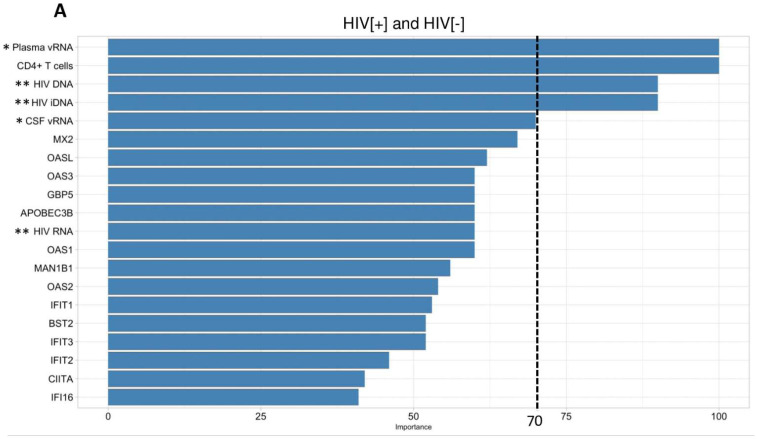
**Analyses of variables predicting clinical group classification by machine learning.** The regularized logistic regression machine learning method was used to predict variables that distinguished the HIV[+] group from the HIV[−] group (**A**). Shrinkage discriminant analysis predicted the variable importance for the HIV[+] and HIV[+]/HAND group classification (**B**). An importance value of 70 was set as a threshold. An importance value between 70 and 100 indicates a heavy reliance of machine learning prediction on a specific variable in classifying and distinguishing the compared groups. (* represents CSF or plasma viral load, and ** represents brain viral load (RNA, DNA or iDNA (integrated DNA).) Host genes were derived from qRT-PCR results.

**Figure 8 viruses-15-00316-f008:**
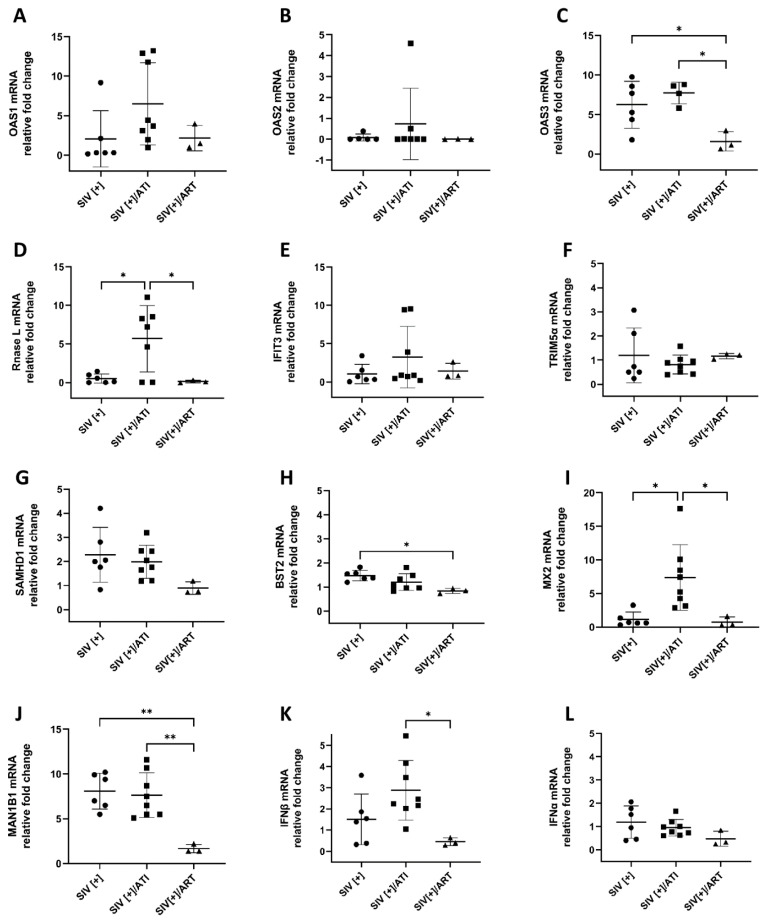
**Analyses of RFs in the brains of SIV-infected non-human primates by qRT-PCR**. The expression levels of 10 known RFs (**A**–**L**), as well as *IFNA* and *IFNB1* were measured in three experimental groups of SIV-infected Chinese rhesus macaques (SIV[+], SIV[+]/ATI and SIV[+]/ART). The mRNA expression levels are normalized to GAPDH and are reported as the fold change relative to the SIV[+]/ART control group. The horizontal lines represent the mean values and the error bars represent the standard deviation *(*, p* < 0.05; **, *p* < 0.01).

**Table 1 viruses-15-00316-t001:** Clinical and demographic features of human cohort.

Groups	Sex	Mean Age (Year) ± SD	Mean Plasma Viral RNA Copies/mL ± SD	Mean CSF Viral RNA Copies/mL ± SD	Average of CD4^+^ T-Cell Cells/µL ± SD	HIV-1 RNA Copies/g of Tissue	HIV-1 DNA Copies/g of Tissue	HIV-1 iDNA Copies/g of Tissue	Neurocognitive Impairment	HIV Encephalitis (HIVE)
**HIV[** **−** **] (*n* = 10)**	9M/1F	48.5 ± 8.1	ND *	ND *	Not available	ND *	ND *	ND *	Not impaired	No HIVE
**HIV[+] (*n* = 10)**	9M/1F	50 ± 5.3	1.33 × 10^5^ ± 338,060	1.62 × 10^4^ ± 1.95 × 10^4^	88.8 ± 95	5.64 × 10^3^	3.57 × 10^4^	8.38 × 10^3^	Not impaired	No HIVE
**HIV[+]/HAND (*n* = 10)**	7M/3F	42 ± 9.5	2.25 × 10^5^ ± 2.47 × 10^5^	1.45 × 10^3^ ± 2.08 × 10^3^	88.2 ± 136	3.59 × 10^4^	4.78 × 10^4^	9.74 × 10^3^	Impaired	No HIVE
**HIV[+]/HIVE (*n* = 10)**	9M/1F	39 ± 5.6	4.44 × 10^5^ ± 3.61 × 10^5^	1.66 × 10^6^ ± 1.24 × 10^6^	47 ± 32	1.05 × 10^5^	1.65 × 10^5^	3.71 × 10^4^	Impaired	With HIVE

* Not Detected.

**Table 2 viruses-15-00316-t002:** Experimental features of SIV-infected non-human primate cohort.

Group	Animal ID	Sex	Age (Year)	Length of Infection (Day)	ART Duration (Days)	ART Interruption (Days)	Plasma Viral Load (Copies/mL) at Death	CSF Viral Load (Copies/mL) at Death	CD3+ CD4+ (%) at Death	SIV RNA Copies/g of Tissue	SIV DNA Copies/g of Tissue	SIV iDNA Copies/g of Tissue
**SIV [+]/No ART (*n* = 7)**	9051222	F	5	35	N/A	N/A	1,580,000.00	15,982.10	34.3	186.25	422,933.33	8620.33
	9082012	F	5	60	N/A	N/A	6340.00	0.00	27.7	0.00	12,920.00	1512.83
	12-2070R	F	6	77	N/A	N/A	755,000.00	445.80	34.8	466.96	2013.33	321.82
	12-1758R	F	8	85	N/A	N/A	36,766.73	596.30	33.90	4281.68	1738.04	533.51
	13-1298R	F	7	307	N/A	N/A	513,557.52	3050.10	20.50	2240.93	1236.66	606.72
	12-1920R	F	7.5	99	N/A	N/A	134.48	194.90	37.10	894.59	135.12	140.08
	13-1572R	F	6.5	104	N/A	N/A	29,486.40	1556.30	24.00	529.40	45.65	540.72
**SIV [+]/ATI (*n* = 8)**	R110482	F	5	69	56	10	110,000.00	0.00	59.0	365.00	27,833.33	4045.38
	R110804	F	5	74	56	15	3710.00	56.10	47.6	216.25	21,740.00	4492.43
	11-1430R	F	5	77	56	18	308,000.00	19,785.60	60.5	0.00	8733.33	1373.62
	12-1888R	F	6	71	56	12	766,000.00	1226.90	60.0	235.38	19,024.44	1496.93
	13-1134R	F	5	74	56	15	34,900,000.00	7816.30	70.3	1511.90	19,151.11	593.02
	13-1878R	F	5	87	56	28	1480.00	0.00	45.7	1694.41	8393.33	606.22
	13-1180R	F	5.5	218	56	159	2634.60	768.70	61.80	221.25	3404.44	446.55
	13-1386R	F	5	220	56	161	18,773.26	256.70	41.70	546.69	8860.00	534.08
**SIV [+]/ART (*n* = 3)**	R110562	F	5	30	27	N/A	0.00	0.00	49.4	1177.50	14,193.33	2333.89
	R110360	F	5	38	35	N/A	0.00	0.00	54.7	0.00	45,406.67	1168.47
	12-1836R	F	6	58	55	N/A	0.00	0.00	41.9	1338.39	2844.44	242.37

## Data Availability

Research data are available and are pending acceptance to NNTC database.

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
