# Peer review of "The HIV Restriction Factor Profile in the Brain Is Associated with the Clinical Status and Viral Quantities"

_viruses, 2023, doi:10.3390/v15020316_

Round 1

Reviewer 1 Report

  • A brief summary The authors are interested in the role of restriction factor expression in the brain of individuals infected with HIV (or SIV) and the link between these observations and HIV-related neuropathy. They use RNAseq, RTqPCR validation and machine learning techniques. They use postmortem brain material from humans and, also, SIV-infected macaques. 
  • General concept comments
    Article: this is an interesting research area and the role of restriction factors in control of viral replication in the brain could have important consequences for understanding viral reservoirs and pathogenesis in infection in HIV and SIV. The experimental methodology appears robust and RNAseq data validated by qPCR. The study is somewhat limited in its scope and some further analysis of existing data may give further insights (see below). 
  • Specific comments 
  • "The differential expression results were screened for 60 previously recognized antiretroviral restriction factor (RF) genes." - provide a reference or methological case for how this list was put together.
  • Figure 2 and S1 - It is difficult to get a sense for the significance of the hits from the presentation of the data. Can the authors plot significance vs fold change for each pairing (volcano plot) in order to highlight particularly interesting hits in each case?
  • Figure 2 - "*indicates gene was selected for". Can the authors clarify what they mean by this statement. 
  • given that the authors have HIV RNA data for the samples, it would strengthen the argument to see any correlation between restriction factor expression and restriction of the virus across samples.

Many thanks

Reviewer 2 Report

Power and colleagues analyzed the expression of antiviral factors in brain sample from HIV-1-infected people with or without neurocognitive disorders and/or encephalitis in comparison to uninfected individuals. They report significant induction of antiviral gene expression that largely correlates with viral loads and provide evidence that MANB1 expression may play a role in HAND. In addition, they examined the expression of restriction factors in brains from SIV-infected rhesus macaques receiving different ART regimens and show that treatment interruption is associated with increased expression of various IFN-stimulated genes (ISGs).

The study addresses an important issue since many HIV-1-infected individuals suffer from neurological symptoms and the brain may also present an import viral reservoir in ART-treated people. The findings that the expression of antiviral factors in the brain positively correlates with the viral loads is not surprising and the potential role of MAN1B1 expression in HAND difficult to assess. Nonetheless, most results seem solid and the study contains interesting information.

Specific points

1. It is still under debate which feature warrant the term “restriction factor”. The authors should briefly describe their definition of restriction factors and on which basis the factors were selected for analysis (also why sometime 30 and other times 60 were chosen).

2. Which genes were analyzed in the NHP cohort? Were the primers used for qRT-PCR analyses adapted to the simian sequences or how was it ensured that they bind effectively to the macaques mRNAs?

Minor points

1. Does “viral burden” refer to viral RNA loads? If yes, I suggest to change the title for accuracy.

2. Mostly well written but some typos should be corrected and some long statements be rephrased or split (e.g. bottom pg. 1: “Although the latent/active…”.

3. Pg. 2 “These antiviral factors have co-evolved with the virus and represent signatures of positive selection.” Rephrase since there are exceptions (e.g. SerinC5).

4. Pg. 6. “HIV[+] (4.1 log RNA copies/ml) (p<0.01) and HIV[+]/HAND (3.1 log RNA copies/ml) (p<0.01)”. It is surprising that the viral load in CSF in the HAND group is substantially lower. This should be discussed and, if possible, explained.

Reviewer 3 Report

This manuscript is a quite interesting study exploring expression levels for a subset of interferon-stimulated genes (ISGs) in the brain of humans and nonhuman primates, infected or not with HIV-1. In short, higher virus loads in the SNC is associated with increased inflammatory symptoms and, accordingly, higher expression levels for ISGs. ISGs are also upregulated in primates following treatment interruption. It is a bit puzzling that the analysis was limited to virus restriction factors instead of ISGs in totality, but perhaps the rest of the data will be published in a future article.

I am not convinced with the final conclusion that “a select group of RFs are induced in the brains of HIV[+] groups with HAND”. Rather, it seems that most ISGs included in the study are found to be upregulated. If the authors wanted to make the demonstration that some specific ISGs were highly upregulated in an unexpected fashion, then maybe they could have tried to correlate the data they obtained with data from studies investigating ISG upregulation in response to inflammatory stimuli other than HIV-1. Nonetheless, these results are unreservedly worthy of publication. Some of my comments below are classified as “Major”, but I believe that only modifications to the text are necessary (no new experiments), and for this reason I evaluate the manuscript as acceptable with minor revisions.

(relatively) Major comments:

1.      A few mRNAs are found to be downregulated in infected humans by RNA-seq, but are upregulated as seen by RT-qPCR. This discrepancy is not explained. Regarding PP1A/CypA, could it be related to the fact that multiple pseudo-genes exist, as well as CypA domains present in other proteins (ex. RanBP2), and perhaps this confuses the RNA-seq data ?

2.      The analysis using machine learning is underwhelming. In particular, it is worrisome that the gene product found to be the most discriminant to classify the various groups is MAN1B1, one of the genes for which there was discrepancy between the RNA-seq and RT-qPCR data. What happens if, instead of the RT-qPCR data, this analysis is done based on the RNA-seq data?

3.      There could be a little more information on the human cohort. What did these persons die of? The low CD4 counts suggest that they died of AIDS. Thus, what is the story of these samples, are these historical samples dating from the pre-ART era that had been frozen for decades prior to this study ? I suppose that this would be fine either way, but it should be made clear to the reader.

Minor comments:

4.      Nucleic acids were extracted from the “midfrontal gyrus” of deceased patients. Explain this choice.

5.      Please verify that all references have the appropriate format. For instance, “Abdel-Mohsen” is written in all-caps

6.      Page 4, “of SIV-infected”: I suppose “macaques” is missing

7.      The title to Fig. 6 could be simpler/shorter, and not include “to validate…”, especially considering that in the end, the results are contradictory.

8.      Discussion, p. 20: there are 2 paragraphs made of a single short sentence each (they start with “HIV Env protein…” and “ERMAN1 besides…”). I think the authors will want to fix this.

9.      In the Acknowledgements, it should be “Université Laval” or “Laval University”, but not “University of Laval”, unless maybe I have been blind to the creation of a new higher-learning institution north of Montreal.  
